# 3T MRI-Radiomic Approach to Predict for Lymph Node Status in Breast Cancer Patients

**DOI:** 10.3390/cancers13092228

**Published:** 2021-05-06

**Authors:** Domiziana Santucci, Eliodoro Faiella, Ermanno Cordelli, Rosa Sicilia, Carlo de Felice, Bruno Beomonte Zobel, Giulio Iannello, Paolo Soda

**Affiliations:** 1Department of Radiology, University of Rome “Campus Bio-medico”, Via Alvaro del Portillo, 21, 00128 Rome, Italy; e.faiella@unicampus.it (E.F.); b.zobel@unicampus.it (B.B.Z.); 2Unit of Computer Systems and Bioinformatics, Department of Engineering, University of Rome “Campus Bio-medico”, Via Alvaro del Portillo, 21, 00128 Rome, Italy; e.cordelli@unicampus.it (E.C.); r.sicilia@unicampus.it (R.S.); g.iannello@unicampus.it (G.I.); p.soda@unicampus.it (P.S.); 3Department of Radiology, University of Rome “Sapienza”, Viale del Policlinico, 155, 00161 Rome, Italy; c.df@uniroma1.it

**Keywords:** breast cancer, 3T-MRI, lymph node status, machine learning, radiomics, signature

## Abstract

**Simple Summary:**

Breast cancer is the most common cancer in women worldwide. The axillary lymph node status is one of the main prognostic factors. Currently, the methods to define the lymph node status are invasive and not without sequelae (from biopsy to lymphadenectomy). Radiomics is a new tool, and highly varied, but with high potential that has already shown excellent results in numerous fields of application. In our study, we have developed a classifier validated on a relatively large number of patients, which is able to predict lymph node status using a combination of patients clinical features, primary breast cancer histological features and radiomics features based on 3 Tesla post contrast—MR images. This approach can accurately select breast cancer patients who may avoid unnecessary biopsy and lymphadenectomy in a non-invasive way.

**Abstract:**

Background: axillary lymph node (LN) status is one of the main breast cancer prognostic factors and it is currently defined by invasive procedures. The aim of this study is to predict LN metastasis combining MRI radiomics features with primary breast tumor histological features and patients’ clinical data. Methods: 99 lesions on pre-treatment contrasted 3T-MRI (DCE). All patients had a histologically proven invasive breast cancer and defined LN status. Patients’ clinical data and tumor histological analysis were previously collected. For each tumor lesion, a semi-automatic segmentation was performed, using the second phase of DCE-MRI. Each segmentation was optimized using a convex-hull algorithm. In addition to the 14 semantics features and a feature ROI volume/convex-hull volume, 242 other quantitative features were extracted. A wrapper selection method selected the 15 most prognostic features (14 quantitative, 1 semantic), used to train the final learning model. The classifier used was the Random Forest. Results: the AUC-classifier was 0.856 (label = positive or negative). The contribution of each feature group was lower performance than the full signature. Conclusions: the combination of patient clinical, histological and radiomics features of primary breast cancer can accurately predict LN status in a non-invasive way.

## 1. Introduction

Breast cancer (BC) is the leading cause of death from cancer in women in Europe [1]. This tumor is considered a pool of different kind of cancer, with various molecular subtypes and with distinct recurrence and survival rates. Among prognostic factors, axillary lymph node status (ALNS) is one of the most important, and its identification is essential for prognosis determination and to guide adjuvant therapy decisions [2,3]. Due to several side effects related to complete dissection, in the last few years, the sentinel node biopsy (SNB) has been proposed as alternative to routine staging for patients with early-stage BC with clinically negative axillary nodes [4,5]. However, also SNB is an invasive procedure and with significant sequelae, including shoulder dysfunction, nerve damage, upper arm numbness, and lymphoedema [5,6]. It remains to be determined whether the emerging evidence for accurate diagnosis and adequate local control with SNB compared with ALN dissection (ALND), is accompanied by equivalent survival outcomes. The ideal would be to predict the ALNS in order to correctly classify breast cancer patients, identifying those patients who will benefit of complete ALND or a more conservative Sentinel Node dissection, avoiding unnecessary invasive treatment [7]. Even more, it is well known that histopathological data of primary tumor, such as lymphovascular invasion, Ki-67 proliferation index, histological grade, estrogen receptor (ER) status and progesterone receptor (PgR) status, are predictors of SLN metastasis. However, they are available postoperatively and cannot be used to guide decisions on performing SLN biopsy [7,8].

In this scenario, MRI has played an evolving role in providing anatomical and functional properties of breast tissues, in a non-invasive way. Findings on MRI such as tumor size, morphology, shape and enhancement have been shown as significant in differentiating breast cancer subtypes [9,10,11,12,13]. In particular, several studies have highlighted the value of dynamic contrast-enhanced MRI (DCE-MRI), identifying it as the main sequence for the detection and characterization of breast lesions, facilitating treatment choice [13,14]. However, manual annotation of tumor characteristics is generally limited to a few qualitative descriptors and is dependent on the operator.

Over the past few years, pilot studies have attempted to correlate the tumor features extracted from MRI with the molecular subtypes of BC. This filed is relatively new and is referred to as radiomics, i.e., the conversion of the information contained in medical images into high-dimensional, mineable and quantitative imaging characteristics, usually referred to as features or descriptors, via high-throughput extraction of data-characterization algorithms [15,16].

A growing interest is expressed in radiomics research in many fields. However, almost all previous works focused on breast cancer, correlated histological and radiomics features of primary tumors and only few works correlate the main breast tumor characteristics with lymph-node status [13,17,18,19,20].

On these premises, the contributions of this work are:To demonstrate that a radiomics-based approach can provide a non-invasive approach for predicting LN metastases useful for clinical practice, which helps identifying those patients who have certain negative lymph node invasion and can avoid unnecessary invasive procedure;To show that the combination of texture features extracted from anatomical and functional DCE-MRI images combined with clinical and histological descriptors boost the performance;To explore a large set of radiomics features, including also a 3D extension of Local Binary Patterns to enrich the description.

The rest of the manuscript is organized as follows: the next section introduces the materials and the methods, and Section 3 presents the results. Section 4 discusses our findings, whilst Section 5 provides concluding remarks.

## 2. Materials and Methods

### 2.1. Patients

All breast MRI exams performed at the Central Radiology Department of Policlinico Umberto I, from January 2017 until January 2019 for pre-operative evaluation, were retrospectively reviewed. The inclusion and exclusion parameters of the study were defined. Only the patients having the following characteristics were included: MRI-examination executed with 3 Tesla magnetic field, post-contrast sequences, mass-like tumors, diagnostic confirmation of invasive breast cancer by histopathological analysis, complete histological analysis including molecular receptor structure and proliferation index Ki67, final lymph-node status (ipsilateral axillary cable).

Patients who had breast implants or expanders, patients in post-chemotherapy follow-up, patients in neo-adjuvant treatment and patients whose images were not of excellent diagnostic quality were excluded. 

For all patients, a written informed consensus was obtained before contrast-MRI execution.

Following the mentioned criteria, a total of 97 breast cancer patients (age range: 37–82 yo; mean age: 55.48 yo) with 99 breast lesions were included in the study (2 bilateral breast cancer cases, 83 invasive ductal carcinoma (83.8%), 12 invasive lobular carcinoma (12.1%) and 4 medullary carcinoma (4.0%).

### 2.2. MRI Examination

All MRI investigations were performed with a 3 Tesla magnetic field with Discovery 750 machinery, from GE Healthcare (Milwaukee, WI, USA), using a breast dedicated 8-channel coil with patient in prone position [11]. Written informed consensus was obtained before each procedure by the patient.

After a sequence performed for framing and carried out along the three space orthogonal planes, were acquired a T2-weighted single shot fast spin eco (IDEAL), a diffusion-weighted sequence (DWI) with b values of 0 and 1000 sec/mm^2^ and T1-weighted 3D axial sequences, dynamic gradient echo, with fat suppression (VIBRANT), before and five times after administration of the intravenous contrast medium (Gadobenate dimeglumine), administered at a concentration of 0.2 mmoL/kg and at a speed of 2 mL/s for a total of 14 mmoL in an ideal patient of 70 kg.

Therefore, “subtracted” images were obtained automatically, using the post-contrast images to which the mask was removed. All the technical details of the sequences are summarized in Table 1.

For each lesion, the following features were collected: localization (breast quadrant position, retro-areolar, upper- or lower-external, upper- or, lower-internal quadrant); margins (divided in regular, irregular, lobulated and spiculated); maximum diameter (mm), measured on post-contrast images; contrast enhancement after contrast agent administration, performed using signal intensity/time curve (type I, with both slow wash-in and wash-out, type II, depicted as a plateau curve and type III, with both rapid wash-in and wash-out).

### 2.3. Clinical Data

Specific anamnestic-clinical data for each patient were previously collected and, according to them, the population was divided into groups: age, menopausal status (42 patients in the pre- and 57 in the postmenopausal phase), hormone therapy (10 patients who have performed at least 3 continuous months of hormone therapy of any kind contraceptive, replacement or therapeutic therapy and 89 patients who did not assume any hormone therapy), familiarity (62 patients without any familiar, 33 patients with one familiar and 4 patients with at least 2 female or male family members affected by breast cancer at any age).

### 2.4. Histological Data

The histological examination was performed for all the 99 breast lesion included in the study. The samples were obtained by core-biopsy or surgery and analysed by an anatomo-pathologist with more than 15 years of experience. The classification of the tumor histotype was performed in accordance with the WHO classification. The tumor histological grade was assigned following the NGS for which a score from 1 to 3 was given for each of these parameters: tubular formation, nuclear pleomorphism and number of mitoses. Immunohistochemical analysis was carried out to determine the receptor structure, Estrogen Receptor (ER), Progesteron Receptor (PgR), Human Epidermal growth factor Receptor (HER2), and the proliferation index Ki67. ER and PgR expression was considered as positive when >10%; Her2 was considered as positive when >+2 and ki67 when >14%.

Hence, the following histological data were collected for each tumor: histotype (divided in invasive ductal carcinoma (IDC), invasive lobular carcinoma (ILC), medullary carcinoma), grading (three groups) and tumor class, basing on hormone receptor expression and proliferation index (Luminal A: ER+, HER2- and low ki67; Luminal B: ER+, HER2 −/+ and high ki67; HER2 overexpressed; Triple Negative (TN): ER−, PgR−, HER2−).

The status of the axilla was assessed after the diagnosis of breast cancer, analysing radiologically (ultrasound of the axillary cable in the diagnostic phase and breast MRI in the staging phase), clinically and then histologically the status of the axillary lymph nodes during definitive surgery. Then, axillary cable definition consisted of sentinel node dissection, sampling dissection or total lymphadenectomy, basing on surgeon decision, but curative in all cases. The patients were simply classified as positive or negative, depending on whether there was, in the first case, at least one lymph node involved, or, in the second case, no positive lymph node. The status of the axillary cable defines the so-called final label, dichotomized into positive LN or negative LN.

In the following sections, we present the method that predicts axillary lymph node metastasis, therefore being a safe and non- invasive prognostic approach. A schema is offered in Figure 1, which consists of 4 blocks, numbered I–IV. According to the figure, next sections first describe the segmentation and the pre-processing approach adopted. Then, we present the proposed method to compute the quantitative descriptors, i.e., feature computation, and next we move to feature selection. Finally, we present the classification approach.

### 2.5. Segmentation and Pre-Processing

This initial step of the method is depicted in the first block of Figure 1. This step also includes data preparation operations, such as the anonymization. The preparation of the images was performed on a personal workstation using 3D Slicer (version 4.8), an open-source software freely available online (http://www.slicer.org, November 2012) [21]. Each case was identified with a progressive identification number (ID). For each tumor, the subtracted post-contrast T1w-MRI was selected. Since five post-contrast phases were available, we used the second sequences, because image lesions in the second phase (60–120 s) had the highest contrast resolution.

At this point, for each case, a label-map was generated. Using manual and assisted segmentation techniques with the thresholding technique, the lesions were manually drawn (Figure 2).

The segmentation was initially always performed in the axial projections and subsequently remodeled and optimized in the other projections until the lesion was contoured optimally, avoiding the necrosis when present into the lesion, as shown in Figure 3.

Any multifocal or multicenter lesions were also segmented. In the event of a bilateral tumors, the lesions were attributed to the same patient but to different IDs, considering them one at a time.

Let us now focus on the segmented slices; we notice that the boundaries of each ROI are often coarse, affecting the quality of the features when calculated. This can be easily understood considering that many of the second-order measures presented in the following are computed from each voxel, considering also the other voxels in its neighbourhood. Before the feature extraction step, we pre-process the lesions’ contours to obtain a 3D volume with smoother edges, thus including large part of voxel neighbourhood. This is performed in two stages. First, we iteratively compute a 3D convex hull (CH), which is the minimum-volume bounded into a convex polygon and containing the ROI [22]; this process begins considering all not connected three-dimensional regions segmented using a default 26-connectivity. Second, all the 3D regions connected using a 26-connectivity are merged into a single volume, determining a new global CH. Such technique iterates until there are no more CHs left to merge, i.e., until the algorithm converges.

The CHs obtained so far are considered the definitive ROIs (Figure 4 depicts an example) and from these volumes we extract the features defined in the next subsection. 

It worth noting the double benefit of applying of the described algorithm. Indeed, it is able to remove all the “outliers” from the ROI, i.e., all sub-volumes that are not large enough to delineate a CH. Moreover, the final volume integrates a slightly larger region surrounding the lesion, thus including all those voxels considered after the conversion of the ROI into a three-dimensional convex shape, and such extra tissue can bring useful quantitative information on how the tumour infiltrates during growth. 

### 2.6. Feature Computation

In addition to the clinical data and the histological data presented in the materials, this work leverages on several radiomic features that can be divided into first-order and texture features. According to Figure 1, this step is performed in the second block. Texture measures, in turn, are derived from the 3D Gray Level Co-occurrence Matrix and from the Three Orthogonal Planes-Local Binary Patterns. Next will be described each feature group.

### 2.7. First-Order Features

They are based on counting of the image voxels grouped by their grey level and, therefore, they measure the intensity density distribution of the ROIs. The literature has shown that humans are mostly sensitive to the light distribution of pixels in images and are better able to discriminate their differences when considering such characteristic rather than others: this is because the first order features are usually referred to as human inspired descriptors. In practice, we extracted the histogram from the 3D ROI and then we computed 12 characteristics up to the fourth-order moments, which are: the mean, the standard deviation, the skewness and the kurtosis. In addition, from the histogram, we also extracted the width, the entropy, the energy, the value of the histogram absolute maximum and the corresponding grey-level value, the energy around such maximum and the number of relative maxima in the histogram and their energy. More details and the definitions of all the features are available in the paper by Cordelli et al. [23].

### 2.8. 3D Gray Level Co-Occurrence Matrix Features 

Usually, healthy tissues and tumours have different textures. Therefore, a deeper focus on the relative distribution of voxels within a ROI rather than simply counting them can reveal many details that can be used as a measure of its microstructure. To this goal, for each ROI we computed the 3D grey level co-occurrence matrix (GLCM3), i.e., a well-known image transformation that captures such a distribution as it can be considered the 3D generalisation of the more popular planar GLCM.

Let us denote with I a 3D greyscale image, and consider a Cartesian reference system O(x, y, z). If we position the origin of the reference system in the upper left corner of I, the position of all its voxels is given by the vector $p=p_{x}\hat{i}+p_{y}\hat{j}+p_{z}\hat{k}$, with $p_{x}$, $p_{y}$ and $p_{z}$ ∈ ℕ, and $\hat{i},\;\hat{j},\;\hat{k}$ are the direction vectors of the axes. We also denote a displacement vector $d=d_{x}\hat{i}+d_{y}\hat{j}+d_{z}\hat{k}$, with $d_{x}$, $d_{y}$ and $d_{z}$ ∈ ℕ. If we define m equal to the number of bit used to represent I, a GLCM3 is a square matrix of size N = 2^m^, where each entry ($g_{i}$, $g_{j}$), with both $g_{i}$ and $g_{j}$ ∈ [0, 2^m^ − 1], represents the number of times a voxel in p with intensity $g_{i}$ is separated by a displacement d from another voxel with intensity $g_{j}$, therefore located in p + d. Denoting as $d_{h}$ the h-th component of d, to assess all possible directions, we considered the combination of $d_{h}$ ∈ {−1, 0, 1} as displacements, without considering the (0, 0, 0) vector yielding to 26 different displacement directions. More details on GLCM3 are reported by Sebastian et al. [24]. As last step, from each GLCM3 we compute 7 measures, namely the autocorrelation, homogeneity, entropy, energy, covariance, inertia and absolute contrast [25]. Concatenating such GLCM3 measures we get 26 × 7 = 182 textural descriptors for each ID. 

### 2.9. Three Orthogonal Planes-Local Binary Patterns Features 

This descriptor is a generalization of well-known planar Local Binary Pattern (LBP) and, although relatively new in radiomics, it can be applied to a three-dimensional image. Now we first presents a brief summary of the original LBP’s concepts and, then, we will introduce the 3D descriptor.

Denoting again a generic 2D image with *I*, if we consider each pixel *p*, it is possible to compare its intensity $I_{p}$ with the intensities of all its j-th neighbours pixels laying on a circle centred in *p* and with radius r. The i-th pixel is set to 1 if $I_{j}>I_{p}$, 0 otherwise. The next step is to process all neighbours of *p* in a circular fashion, read the resulting sequence of 0 s and 1 s as a binary string and encode the value of *p* to the equivalent decimal value. In practice, all the pixels in *I* are processed following this procedure to obtain an image encoding all the intensity distributions of the pixels with respect to their neighbours. Using this descriptor, we thus grasp part of textural information contained in *I*. Finally, notice that, if we denote with P the number of local neighbours that surround the central point *p*, the number of patterns for this planar LBP implementation is 2^P^.

In order to extend this approach to a three-dimensional environment, in [26] the authors presented a solution that considers an helicoidal neighbourhood for each voxel. However, this produces 2^3P+2^ patterns; to avoid this computational burden, we introduce another 3D implementation of LBP transformation that considers the co-occurrence on three orthogonal planes crossing the center of the analysed volume, as detailed in [26]. This method is named Three-Orthogonal-Planes LBP (TOP-LBP) and computes three 2D LBPs, one for each plane, and the obtained histograms are concatenated to obtain a unique representation for the specific volume. This conspicuously alleviates the computational load, since the number of patterns for TOP-LBP is 3 × 2^P^. Furthermore, in our LBP implementation, we consider two more variants to cope with other two issues of 2D LBP definition. First, we computed rotation invariant LBP, i.e., all binary strings obtained as the circular shift of a fundamental string are considered the same. Second, we implemented a uniform version of LBP, i.e., all binary strings containing more than two crossings from 0 to 1 or from 1 to 0 are considered not uniform and coded with a specific string. In our case, setting P = 8, we get 48 features by computing first-order measures from each histogram of the three 2D LBP. 

### 2.10. Feature Selection

One of the main goals of radiomics is to find a signature, i.e., the set of all those features with the highest discriminative power for the task at hand. Moreover, this is also beneficial for the learning phase of a classification algorithm, as reducing the number of features to be considered reduces the risk of the curse of dimensionality. To this goal, we searched for the most discriminative features by using a wrapper (third block of Figure 1). This feature selection method is known for being able to find dependencies between the descriptors and, at the same time, it exploits the interaction between the subset of features to be found and the model selection itself [27]. In fact, after determining a search method for all possible subsets of features, it evaluates them by training and then testing a specific classification model.

We used the Random Forest (RF) as a learning paradigm evaluating the feature subsets by maximizing the area under the curve of the receiving operator (AUC-ROC). The reason for selecting the RF is its ability to work with both qualitative and quantitative features, its capacity to reduce the risk of data overfitting, and the fact that it can also handle datasets with a large number of characteristics. Furthermore, for all its parameters, we used the default values provided by the Matlab library, without any fine tuning. Indeed, we were not interested in the best absolute performance and, also, it has been empirically observed that in many cases the use of tuned parameters cannot significantly outperform the default values of a classifier suggested in the literature [28].

With regard to the subset exploration strategy, it is worth noting that we set the wrapper to work with the best first search by proceeding with nested cross-validation to evaluate the model. This approach ensures that the performance of the model during validation is not affected by a possible favourable split of the data into training and testing, and thus eliminates any bias in the final performance evaluation since the outer test set was never used in the wrapper model evaluation, as also shown in the third block in Figure 1. 

In practice, in each cycle of external cross-validation all samples are divided into a training set and a test set. Then, in the feature selection we take into account only the training samples and apply an additional five-fold internal cross-validation to them. On each inner loop, we test several subgroups of *Fm* features, where each subgroup is composed of the first *m* features of the whole pool. 

Finally, for constructing the radiomic signature, we consider only those features that are selected at least 20% of times in the nested cross-validation experiments described so far. The purpose of this approach is two-fold: first, in this way a balance is maintained between the discriminative power of *Fm* and the risk of overfitting, also reducing the curse of dimensionality which, although mitigated thanks to the use of RF, still remains partly present; second, the validity of this method has been experimentally determined in some preliminary tests, omitted here in order not to burden the discussion. In conclusion, this procedure returned the final set of 15 features shown in Table 2. 

### 2.11. Classification and Model Construction

The fourth and last block in Figure 1 depicts the final classification stage, which provides the resulting performance when using a RF as classifier. As before, we set its parameters to the default values. In order to cope with the class imbalance in the dataset we introduce a misclassification cost making the algorithm cost-sensitive. Let us recall that cost sensitive learning is one of the three main approaches that can be used to address class imbalance, a.k.a. class skew, which is the problem of having a disproportionate training set among different classes. This issue arises since traditional learning algorithms are designed to minimize errors over the majority samples, ignoring or paying less attention to instances of the minority classes, and this usually results in poor predictive accuracy over the minority ones. The predicted label will assume values of 0 in case of negative axillary involvement, and 1 in case of positive axillary involvement. On this basis, in the following, we consider an error cost matrix that sets to 0.59 the cost of false negative errors (FN error = 0.59) and we set to 0.41 the cost of wrongly classifying a negative lymph node into a positive one (FP error = 0.41). Straightforwardly, no penalties are set in the case of a correct prediction. The rationale lies in observing that improperly indicating the presence of a LN metastasis is less dangerous than wrongly indicating its absence, since in the latter case the patient will be no further treated; we account for this by setting misclassification cost to 50% more than the value of the other one.

We performed all the experiments in ten-fold cross-validation, using different randomly generated partitions when cycling over the inner loop with respect to the outer cross-validation loop of the feature selection. This approach as well as not using the outer test partition in the feature selection stage, avoids any bias into the final classification model.

Finally, in order to quantify the results obtained in all the outer folds and, therefore, evaluate the overall classification performances, we built the ROC and computed the underlying area (AUC).

## 3. Results

In this study, 97 breast cancer patients with 99 breast lesions were enrolled (2 patients had bilateral breast cancer). The average age of the entire cohort was 55.48 years. There were 27/99 axillary cables confirmed as ALN positive metastases and the rest 72/99 as negative.

The clinical and pathological characteristics of the patients are summarized in Table 3.

For the sake of presentation, the proposed radiomic approach is divided into specific steps as shown in Figure 1. After the feature selection, a signature of 15 features that lead to the final results was individuated. The selected features were 14 radiomic features and one semantic.

These features were used to build the classifier, whose performance is shown in the first row of Table 4. Our classifier achieved a sensibility and specificity of 86% and 74%, respectively. To deepen the results, we perform additional experiments by investigating one group of features at a time. Hence, we obtained four features groups (1° order, GLCM3, LBP-TOP and semantic features) in addition to the signature used to build the proposed classifier. The sensibility and specificity calculated by each feature group were 85% and 48% for 1° order features, 79% and 52% for GLCM3 features, 69% and 41% for LBP-TOP features and 97% and 11% for semantic features, respectively.

The accuracy and the AUC obtained by the proposed method and by each features class considered are reported in Table 4. Furthermore, Figure 5 shows the ROC curve. 

## 4. Discussion

In the literature, it is well known that the condition of the axillary lymph nodes in breast cancer patients is one of the most important prognostic factors in terms of loco-regional recurrence and overall survival. In this regard, the American College of Surgeons conducted an important study on 20,547 patients with local or regional disease treated with radical mastectomy, and identified a recurrence rate of 19% and 49% and 5-year survival rates without recurrence of 60% and 35%, for patients with negative and positive lymph nodes, respectively [29]. Given the importance of lymph nodes status, not only for the prognosis but also in the guide of oncological therapeutic choice, safer and non-invasive approaches have been investigated for the initial lymph node staging [3,8,20].

Today, we know that radiological images contain much more information than that which is perceivable and visible to the radiologist. This hidden information can provide several interesting data about the tissues, data that are quantifiable. As stated by Gillies RJ et al. in [15], radiomics is based on this principle because it extracts and analyses large quantities of characteristics, defined as “features”, computed from medical images routinely acquired.

In this work, we present an approach for providing valid, rapid and non-invasive support in the prediction of axillary lymph node status, using radiomics features computed from post-contrast MR images associated with patient clinical information and tumor histology.

To date, there are works on breast radiomics which evaluate its effectiveness in diagnosis, identification or characterization, prognosis or response to therapy, using the imaging information produced by different techniques (US, mammography and MRI). The most recent review on radiomics and breast cancer [16] has only 17 studies. Additionally, even fewer are the studies which evaluate the use of breast radiomics in predicting axillary lymph node status. To the best of our knowledge, there are six main papers that have been published since 2017. All are monocentric retrospective studies, with a population ranged from 62 to 163 patients with only one work using 411 samples. Our study includes 97 patients with 99 breast lesions, being in the average of the current literary trend. Chai et al. analyse the features with and without contrast medium agent sequences and compare the two derived accuracy and AUC [30], while Dong et al. use only the T2-weighted and diffusion sequences with the aim of identifying an instrument that prevents the use of the intravenous contrast medium, obtaining AUC values of 0.770 and 0.787, respectively [19]. Our contribution and all the other work in the literature [13,17,18,20,30] use only contrast enhanced sequences: this choice is motivated by the results obtained from studies of both breast radiomics [16,31] and breast MRI [13,14], which identify in the contrast enhanced sequence the main sequence for detection and characterization of the breast lesion.

Regarding the choice of the contrast enhanced phase, there is currently no consensus in defining in which phase the extraction of features offers the best forecast. Liu et al. have applied, for the same purpose of this paper, radiomic features extracted from the first contrast enhanced phase obtaining an AUC of 0.806 [17]. In our study, instead, the second contrast enhanced phase was used.

The higher contrast resolution of the contrast enhanced sequences among all the sequences allows a high definition of the morphological lesion details [14]. Even more, the use of a high-field magnet, 3 Tesla here, guarantees the production of high temporal and spatial resolution image [32]. Obtaining high quality images not only allows a more adequate definition and selection of the features, but reduces the classifier noise.

In our sample of 99 breast tumors, the selection method applied identified 14 quantitative radiomic and 1 semantic features, represented by tumor histological class, correlating significantly with the lymph node status.

Among the 15 selected features, the most numerous were first and second order ones, which describe the intensity and textural characteristics of the tumor, representing the intratumoral heterogeneity and the subtle alterations of the morphology of the tissues. The selection of tumor histological class and not grading or dimension, as we can it can be expected, may be explained to the type of wrapper used for the selection. The reason of such behaviour is probably relied to the multivariate approach of the chosen feature selection algorithm. In fact, when considering multiple features pools, the discrimination power of each tested subset is usually outperforming the capability of predicting the correct class using single features. Therefore, when considering a large set of features mainly computed from the images, where only a smaller percent belongs to a semantic nature (i.e., the clinical and histological data), the final signature is reasonably expected to contain only those semantic features useful to maximize the performance when used in conjunction with the radiomics features. Another motivation is related to the physiologic tumor behavior since, as demonstrated by other studies [30], ductal carcinoma is characterized by morphological and enhancement patterns which are expression of more aggressive behaviour.

Then, the combination of the selected features made it possible to train a classifier capable of predicting the lymph node status in patients with breast cancer, with an AUC of 0.86. The AUC reported in Figure 5 shows the relation between the true positive (sensitivity) and the false positive (1–specificity) rates. Our classifier identifies 20/27 TN patients with a specificity of 74%, which is considered a very good result in comparison with US overall specificity for palpable and non-palpable LN, which it is based on the LN size criteria and it ranges between 44.1% and 97.9% [33]. It is interesting to note that the specificity achieved using only semantic feature is of 11%, indicating the relevance of radiomics as supporting tool.

An example of how the classifier correctly works is reported in the two cases shown in Figure 6.

The study has several aspects worth highlighting. The set of analysed images was acquired in a single center using a single MRI device, repeating the same protocol for all patients, which allows maximum reproducibility in the extraction and analysis of radiomic characteristics. Segmentation was performed for all patients by a single radiologist, who used the same methodology; also the extraction of the features was performed starting from free commercial software, validated in many previous studies, but modified according to the specific study needs, in particular the features analysed included also a 3D extension of Local Binary Patterns in addition to the best known first-order features, GLCM and LBP.

There are also several limitations: first of all, it is a retrospective study with a relatively small sample. The provenance of the population from a single medical center limits the possible generalization and results. A larger cohort of patients, from multiple centers, is needed for a more rigorous analysis. Secondly, ROIs were drawn manually by the radiologist, so they are time-consuming and subject to user errors and variability. An automatic, reliable and validated segmentation method is ideal but not yet available. Thirdly, it was decided to use only the contrast enhanced sequences. Although these have been demonstrated to have the best accuracy, we recognize that it could be limiting and, in any case, lose some important information provided by the other sequences.

## 5. Conclusions

Our results suggest that histological data and radiomics features can be combined for the prediction of lymph node metastases, guiding the treatment planning. The results achieved suggest that they have the potential to impact the clinical practice by offering to clinicians and to patients the possibility of avoiding invasive procedures such as lymphadenectomy or lymph node biopsy in unnecessary cases.

## Figures and Tables

**Figure 1 cancers-13-02228-f001:**
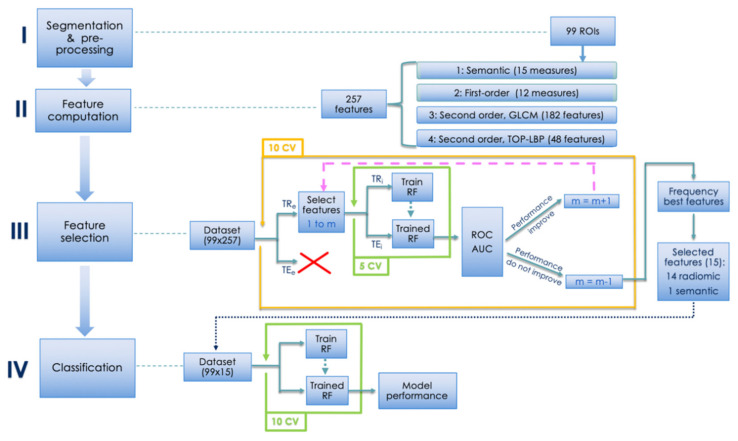
Schematic representation of the proposed approach. TR and TE denote the training and the test set, whilst the subscript *_e_* and *_i_* stand for external and internal cross-validation, respectively.

**Figure 2 cancers-13-02228-f002:**
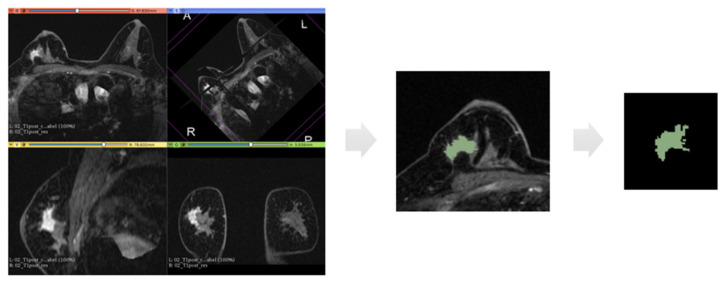
Representation of the extraction of a segmentation mask. From left to right: first panel represents an example of 3D image analysed in our dataset which is the second phase of subtracted post-contrast sequence; then, for each 2D slice, we have the region of interest drawn around the tumour mass and finally the binary mask extracted from this segmentation.

**Figure 3 cancers-13-02228-f003:**
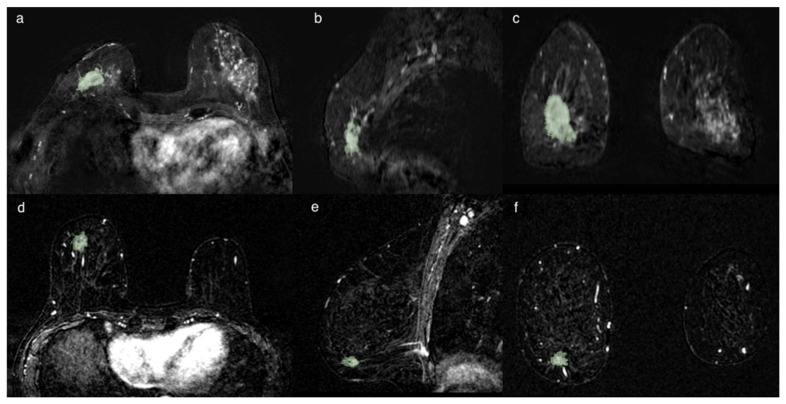
Two cases of breast cancer with positive and negative lymph node axillary involvement at definitive surgery, respectively. The first (**a**–**c**) represented by a 32 mm mass, in 56yo patient with TN CDI tumor characterized by spiculated margins, heterogeneous enhancement with a necrotic core and a signal/intensity curve type 3 at MRI exam. The second (**d**–**f**) is a 13 mm nodule, in a 61 yo patient, LUMINAL A CDI, characterized by irregular margins, heterogeneous enhancement and a signal/intensity curve type 3. The segmentation has been performed in the axial image (**a**,**d**), following the margins and including the spicule characterized by contrast-enhancement. The segmentation was then optimized in the sagittal (**b**,**e**) and coronal (**c**,**f**) planes, avoiding the darker part representing the necrosis and the vessels.

**Figure 4 cancers-13-02228-f004:**
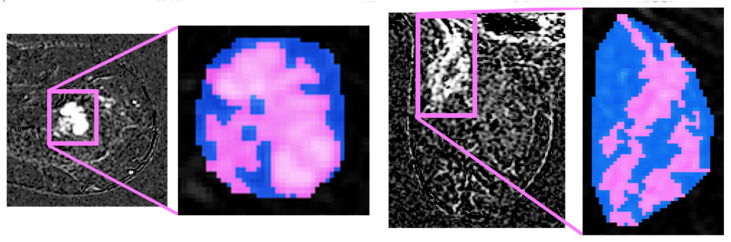
Four pre-processing examples. Left side halves: Bifocal lesion localized at upper-external quadrant, characterized by spiculated margins and IS/T curve. Right side halves: in pink, we highlight the original segmentation and, in blue, the convex hulls obtained.

**Figure 5 cancers-13-02228-f005:**
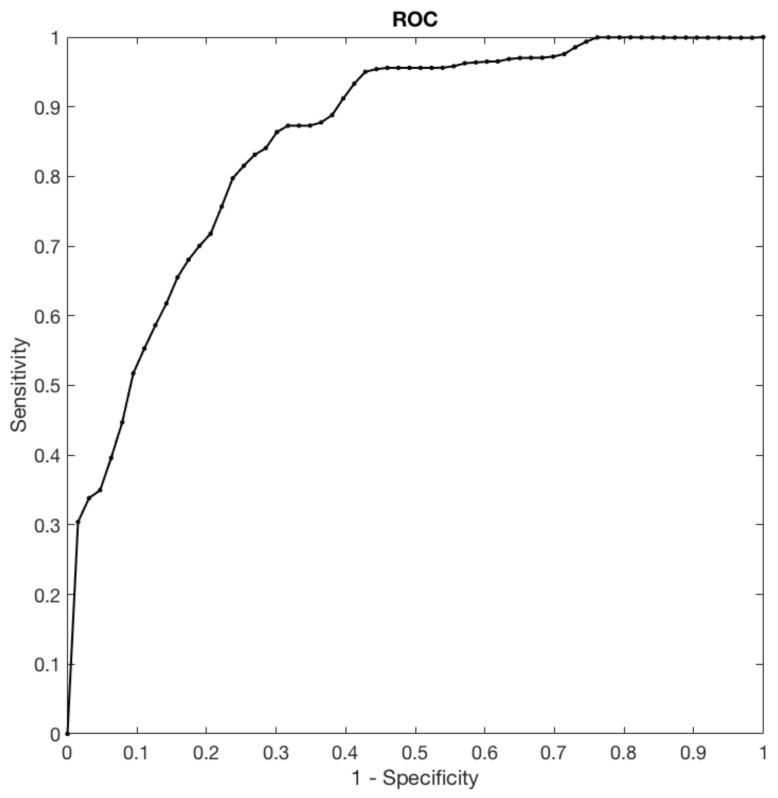
The chart shows the ROC curve which is representative of the classifier performance.

**Figure 6 cancers-13-02228-f006:**
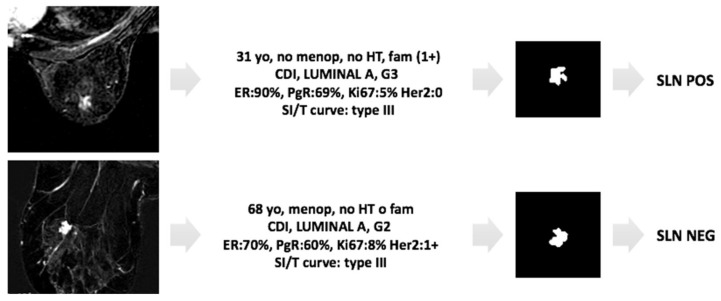
Example of applicability of the proposed approach. Two cases of invasive breast cancer, characterized by very similar MRI features (morphology, margin, post-contrast intensity, IS/T curve, and size and position of the two lesions are comparable) as shown in the left panels. The first case (first row) is a ductal G3 LUMINAL A carcinoma, in 31yo patient with familiarity, the second one (second row) is also a ductal G2 LUMINAL A carcinoma, but in 68yo patient without familiarity. Despite the common MRI and histological features, the produced classifier correctly categorizes the two patients as positive and negative LN status, respectively.

**Table 1 cancers-13-02228-t001:** Study protocol adopted for MRI examination.

Sequences	Values
Axial T2 IDEAL FSE	
TR/TE (ms)	3500–5200/120–135
Matrix	352 × 224
FoV (mm)	370 × 370
NEX	1
Slice thickness (mm)	3.5
Axial EPI DW	
TR/TE (ms)	2700/58
Matrix	100 × 120
FoV (mm)	360 × 360
NEX	6
B-values (s/mm^2^)	0–1000
Slice thickness (mm)	5
Axial T1 VIBRANT	
TR/TE (ms)	6.6/4.3
Angle	10°
FoV (mm)	380 × 380
Matrix	512 × 256
NEX	1
Slice thickness (mm)	2.4

TE: Time of Echo. TR: Time of Repetition. FOV: Field of View. FSE: Fast Spin Echo. EPI DW: Echo Planar Imaging Diffusion-Weighted. VIBRANT: Volume Imaging for BReast AssessmeNT.

**Table 2 cancers-13-02228-t002:** The final feature set.

Histotype	Semantic
Energy around relative maxima	1st order
Energy around absolute maximum	1st order
Energy in direction (−1.1.0)	GLCM3
Energy in direction (0.−1.1)	GLCM3
Energy in direction (0.−1.−1)	GLCM3
Energy in direction (0.−1.1)	GLCM3
Inverse in direction (−1.0.0)	GLCM3
Inverse in direction (1.−1.1)	GLCM3
Inverse in direction (−1.0.−1)	GLCM3
Range	LBP-TOP
Range U	LBP-TOP
Range U RI	LBP-TOP
Number of relative maxima RI	LBP-TOP
Energy around relative maxima RI	LBP-TOP

U—uniform. RI—rotation invariant.

**Table 3 cancers-13-02228-t003:** Main frequencies for the various semantic classes analyzed.

Class	Group	Frequency	Percentage
Familiarity	none	62	62.6
	1 fam	33	33.3
	>1 fam	4	4.0
HT	no	89	89.9
	yes	10	10.1
Menopause	no	42	42.4
	yes	57	57.6
IS curve/T	I	3	3.0
	II	53	53.5
	III	43	43.4
Margins	regular	3	3.0
	irregular	46	46.5
	lobulated	30	30.3
	spiculated	20	20.2
Histotype	IDC	83	83.8
	ILC	12	12.1
	Medullary	4	4.0
Grading	1	12	12.1
	2	45	45.5
	3	42	42.4
Class	Luminal A	39	39.4
	Luminal B	35	35.4
	Her2	9	9.1
	TN	16	16.2

**Table 4 cancers-13-02228-t004:** Metrics that describe the various accuracies considering each feature class stand alone and the proposed approach, obtained by the combination of all the features classes.

Features	TP (%)	FP (%)	TN (%)	FN (%)	Accuracy	ROC Area
Proposed	62 (89)	7 (11)	20 (67)	10 (33)	0.828 (95% CI, 73.6% to 92.0%)	0.856 (95% CI, 77.5% to 93.7%)
Clinical	71 (75)	24 (25)	3 (75)	1 (25)	0.748 (95% CI, 70.2% to 79.4%)	0.533 (95% CI, 45.7% to 60.9%)
First order	61 (81)	14 (19)	13 (54)	11 (46)	0.748 (95% CI, 66.6% to 83.0%)	0.665 (95% CI, 58.4% to 74.6%)
GLCM3	57 (83)	13 (17)	14 (48)	15 (52)	0.717 (95% CI, 64.1% to 79.3%)	0.763 (95% CI, 65.4% to 87.2%)
LBP-TOP	50 (76)	16 (24)	11 (33)	22 (67)	0.616 (95% CI, 54.2% to 69.0%)	0.618 (95% CI, 53.0% to 70.6%)

Where TP = True Positive. FP = False Positive. TN = True Negative. FN = False Negative. ROC = Receiver Operating Curve. CI = Confidence Interval.

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
