# Peer review of "3T MRI-Radiomic Approach to Predict for Lymph Node Status in Breast Cancer Patients"

_cancers, 2021, doi:10.3390/cancers13092228_

Round 1

Reviewer 1 Report

The approach, described in the present work,  based on accurately selection of breast cancer patients avoiding unnecessary biopsy and lymphadenectomy in a non-invasive way, is very appreciable. In their results, the authors clearly underline how the combination of patient clinical, histological and radiomics features of primary breast cancer can accurately predict LN status in a non-invasive way.

Some minor revisions are necessary, mainly about materials and methods. The author could refine the description of them.

- In section 2.1 of Materials and Methods, it may be useful to add more details of study patients, even if there are in results paragraph: number of patients, average age, characteristics of breast lesions

- In section 2.2 of Materials and Methods, it is necessary a revision of administered concentration of CA, adding the value of administered total volume.

- In section 2.3 of Materials and Methods, the authors could be indicate the number of patients of each group in which the population was divided.

- In section 2.4 of Materials and Methods, how many histological data have analyzed ? and how many for each tumor histological grade.

Author Response

Reviewer 1

The approach, described in the present work,  based on accurately selection of breast cancer patients avoiding unnecessary biopsy and lymphadenectomy in a non-invasive way, is very appreciable. In their results, the authors clearly underline how the combination of patient clinical, histological and radiomics features of primary breast cancer can accurately predict LN status in a non-invasive way.

Some minor revisions are necessary, mainly about materials and methods. The author could refine the description of them.

- In section 2.1 of Materials and Methods, it may be useful to add more details of study patients, even if there are in results paragraph: number of patients, average age, characteristics of breast lesions

We thank the Reviewer for the suggestion, and the following sentence is added to the revised version of the manuscript: “Following the mentioned criteria, a total of 97 breast cancer patients (age range: 37-82 yo; mean age: 55.48 yo) with 99 breast lesions were included in the study (2 bilateral breast cancer cases, 83 invasive ductal carcinoma (83.8%), 12 invasive lobular carcinoma (12.1%) and 4 medullary carcinoma (4.0%)).”

- In section 2.2 of Materials and Methods, it is necessary a revision of administered concentration of CA, adding the value of administered total volume.

We thank the Reviewer for the suggestion, and the following sentence is modified in the revised version of the manuscript: “concentration of 0.2 mmol/kg and at a speed of 2 ml/sec for a total of 14mmol in a ideal patient of 70kg”.

- In section 2.3 of Materials and Methods, the authors could be indicate the number of patients of each group in which the population was divided.

We thank the Reviewer for the suggestion, and in the revised version of the manuscript we specify the requested data as follows: “Specific anamnestic-clinical data for each patient were previously collected and according to them, the population was divided into groups: age, menopausal status (42 patients in the pre- and 57 in the postmenopausal phase), hormone therapy (10 patients who have performed at least 3 continuous months of hormone therapy of any kind contraceptive, replacement or therapeutic therapy and 89 patients who did not assume any hormone therapy), familiarity (62 patients without any familiar, 33 patients with one familiar and 4 patients with at least 2 female or male family members affected by breast cancer at any age).”

- In section 2.4 of Materials and Methods, how many histological data have analyzed ? and how many for each tumor histological grade.

We thank the Reviewer for the observation, and in the revised version of the manuscript we specify the requested data as follows: “The histological examination was performed for all the 99 breast lesion included in the study.” “ER and PgR expression was considered as positive when >10%; Her2 was considered as positive when >+2 and ki67 when >14%. Hence the following histological data were collected for each tumor: histotype (divided in invasive ductal carcinoma (IDC), invasive lobular carcinoma (ILC), medullary carcinoma), grading (three groups) and tumor class, basing on hormone receptor expression and proliferation index (Luminal A: ER+, HER2- and low ki67; Luminal B: ER+, HER2 -/+ and high ki67; HER2 overexpressed; Triple Negative (TN): ER-, PgR-, HER2 -).”

Reviewer 2 Report

The authors  have developed a 3T MRI-Radiomic approach to predict for lymph node status in breast cancer patients.

The method is well decribed.

Please clarify how the lymph node assessment was performed: did each  patient received an axilla dissection? If not, how can you exclude positive nodes in MRI negative lesions?

Author Response

Reviewer 2

The authors  have developed a 3T MRI-Radiomic approach to predict for lymph node status in breast cancer patients.

The method is well decribed.

Please clarify how the lymph node assessment was performed: did each  patient received an axilla dissection? If not, how can you exclude positive nodes in MRI negative lesions?

We thank the Reviewer for the observation that permits us to enrich the corresponding original sentence with the following details (Section 2.4): “The status of the axilla was assessed after the diagnosis of breast cancer, analysing radiologically (ultrasound of the axillary cable in the diagnostic phase and breast MRI in the staging phase), clinically and then histologically the status of the axillary lymph nodes during definitive surgery. Then, axillary cable definition consisted of sentinel node dissection, sampling dissection or total lymphadenectomy, basing on surgeon decision, but curative in all cases”

Reviewer 3 Report

Dear Authors,

you wrote: 

Radiomics MRI for lymph node status prediction in breast cancer patients: the state of art

Calabrese, A., Santucci, D., Landi, R. et al. Radiomics MRI for lymph node status prediction in breast cancer patients: the state of art. J Cancer Res Clin Oncol (2021). https://doi.org/10.1007/s00432-021-03606-6

and the manuscript submitted to "CANCERS" is very close to the research published on J Cancer Res Clin Oncol.

Thank you

Author Response

Reviewer 3

Dear Authors,

you wrote: 

Radiomics MRI for lymph node status prediction in breast cancer patients: the state of art

Calabrese, A., Santucci, D., Landi, R. et al. Radiomics MRI for lymph node status prediction in breast cancer patients: the state of art. J Cancer Res Clin Oncol (2021). https://doi.org/10.1007/s00432-021-03606-6

and the manuscript submitted to "CANCERS" is very close to the research published on J Cancer Res Clin Oncol.

Thank you

The article, entitled “Radiomics MRI for lymph node status prediction in breast cancer patients: the state of art, Calabrese, A., Santucci, D., Landi, R. et al. Radiomics MRI for lymph node status prediction in breast cancer patients: the state of art. J Cancer Res Clin Oncol (2021). https://doi.org/10.1007/s00432-021-03606-6", and mentioned by the reviewer, is a review of literature data about the topic Radiomics and lymph node status in breast cancer. This research has been conducted in order to resume all the current literature publishing a state of the art of this topic, before to proceed with an own study.
The manuscript submitted to Cancers is, instead, an original research and it reports a specific experience conducted by our research group in our center.

Round 2

Reviewer 3 Report

none